# FedTED: Federated Learning for Robust Thyroid Eye Disease Detection with Masked Autoencoders

Wai Tak Lau[1], Angela McCarthy[2], Ye Tian[3], Christopher Nielsen[4], Rashmi Chelliah[1], Sina Gholami[5]
Andrea Kossler[4], Minhaj Alam[5], Lora Dagi Glass[6], Kaveri A. Thakoor[1,3,6]

Departments of [1]Computer Science and [3]Biomedical Engineering, Columbia University, New York, NY, USA
[2]University of Connecticut School of Medicine, Farmington, Connecticut, USA
[4]Department of Ophthalmology, Byers Eye Institute, Stanford University School of Medicine, Palo Alto, CA, USA
[5]Department of Electrical and Computer Engineering, University of North Carolina, Charlotte, NC, USA
[6]Department of Ophthalmology, Columbia University Irving Medical Center, New York, NY, USA

*Abstract*—**Thyroid eye disease (TED) detection presents diagnostic challenges due to its heterogenous clinical presentation and limited data availability across institutions. In this paper, we propose FedTED, a privacy-preserving framework that integrates Federated Learning and self-supervised pretraining to detect TED from external facial images without sharing sensitive patient data. Our method integrates masked autoencoders to capture rich representations, followed by fine-tuning under a federated setting. We evaluate FedTED across different training regimes and show that federated MAE-based models outperform supervised baselines, achieving highest performance - AUC up to $98.70\%$ - across validation folds. These results demonstrate the feasibility and utility of combining federated learning with self-supervised training for sensitive medical applications, particularly in settings with limited data and privacy constraints. In clinical settings, this translates to potential for deploying robust models in the real world for early disease detection.**

*Index Terms*—**Federated Learning, Self-Supervised Training, Thyroid Eye Disease**

## I. INTRODUCTION

Thyroid eye disease (TED), otherwise known as Graves Ophthalmopathy or Graves Orbitopathy, is an autoimmune disorder caused by the same antibodies as autoimmune thyroid disorders. In TED, multiple different parts of a patient's immune system are inappropriately called to action, causing inflammation, swelling, excess tissue, and scarring [1]. The majority of patients with TED have hyperactive Graves' disease of the thyroid, though it is also possible to have an underproductive or even a normal thyroid level. Diagnosis of TED is classically based on the marriage of clinical appearance and labwork demonstrating the presence of thyroid antibodies [2], though radiologic imaging can be helpful in confirming the diagnosis and/or preparing for surgery. The clinical appearance of TED can vary greatly between individuals, and may include retraction, or extra stimulation, of the eyelids, known as a 'thyroid stare;' bulging of the eyes, known as exophthalmos or proptosis; cross-eyed appearance, known as strabismus; and swelling and/or redness of the skin and tissues around and of the eye. TED can be graded as mild, moderate, or severe in nature, and in severe TED patients are at risk of losing vision due to compression of the optic nerve, which brings visual data from the eye to the brain. TED typically stabilizes over several years, by which time many patients may look permanently different than their pre-TED state.

Unfortunately, surgery is an imperfect medium, and current treatment and research is aimed at considering medical treatments that may be started earlier in the disease process in order to help permanently prevent some or, ideally, all of the changes of TED. Currently, patients are referred to TED experts (typically oculofacial plastic and reconstructive surgeons) by endocrinologists, primary care physicians, and ophthalmologists who note possible evidence of TED. However, this referral pattern typically means patients have significant enough TED to be obvious to the relatively untrained eye. Additionally, while clinical trials currently aim to identify patients with moderate TED – bad enough to be on a trial, but not so severe so as to preclude immediate treatment with known agents – the future of TED is one in which treatments will be gentle and effective enough to start in the earliest, most mild cases so as to preclude moderate and severe disease. Therefore, creating an automated system to easily and quickly identify TED of all severity types would be invaluable in both current and future states of TED treatment.

AI, particularly deep learning, has shown promise in detecting TED from orbital CT/MRI and external facial photos. However, developing robust models requires large, diverse datasets, especially for identifying subtle or early-stage disease. This is challenging because facial images are highly sensitive and cannot be freely shared between institutions, and because such images are not routinely collected from patients without TED, limiting the pool of eligible control cases. Given the relatively low prevalence of TED at a single institution [3], [4], assembling a sufficiently large and representative dataset is rarely feasible at one site alone. Federated learning offers an ethical solution by enabling multi-institutional AI development without requiring the transfer of facial images, thus improving model diversity and generalizability.

We propose a federated learning framework that integrates self-supervised pretraining, enabling collaboration across different institutions while preserving data privacy. This approach increases the diversity of TED manifestations represented in the training set, thereby improving the model robustness. Additionally, self-supervised pretraining improves representa-

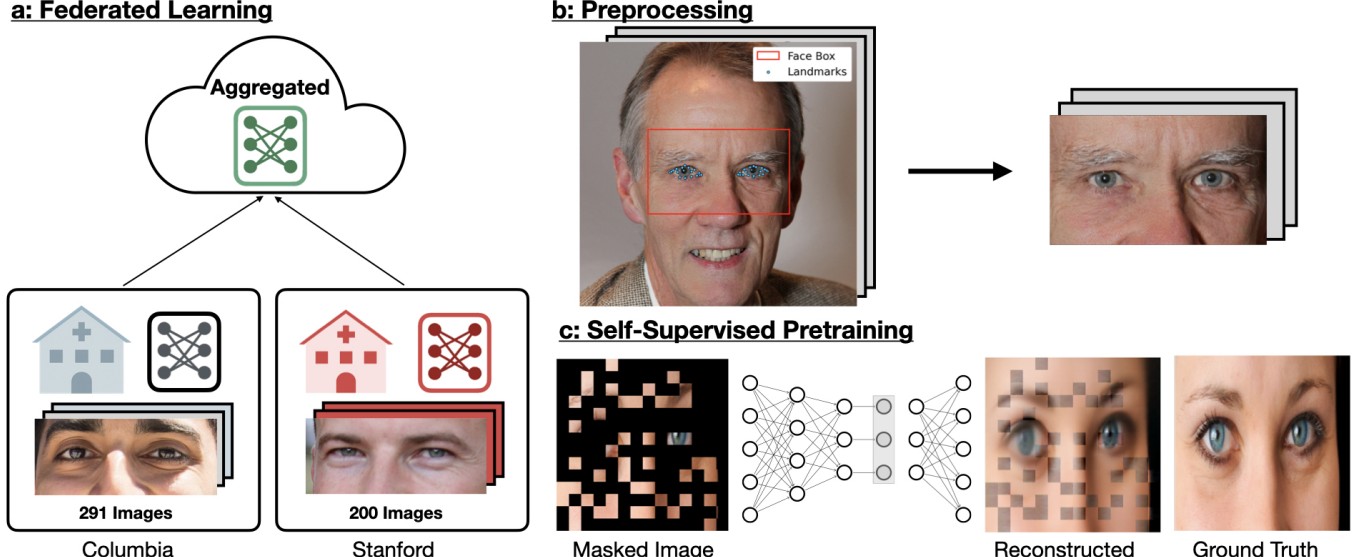

**a: Federated Learning**

Aggregated

291 Images
Columbia

200 Images
Stanford

**b: Preprocessing**

Face Box
Landmarks

**c: Self-Supervised Pretraining**

Masked Image    Reconstructed    Ground Truth

Fig. 1: **a**: FL preserves privacy by repeatedly aggregating models trained on clients' data. Different colors indicate that data remains local, with each site and the aggregator maintaining its own model copy: blue for Columbia, red for Stanford and green for the aggregated global model. **b**: Preprocessing helps to standardize inputs. We first detect key points and then crop the periocular region. **c**: Self-supervised pretraining helps to improve representations. Figure presents MAE; contrastive methods are also compared in our study. Patches of training images are masked as a pretext task, enabling the encoder to learn meaningful features for reconstruction. Images shown are from [5].

tion quality, which is particularly important for fine-grained classification.

Our contributions are as follows:

- We propose FedTED, the first framework that combines federated learning with masked autoencoder pretraining for Thyroid Eye Disease detection from external facial images. This enables cross-institution collaboration without sharing sensitive data.
- We conduct extensive experimentation across clinical sites to show the utility of personalized vs. global federated learning and self-supervised (SSL) pretraining, outperforming local and non-SSL-pretrained baselines in classification performance.
- We systematically compare different widely adopted pretraining strategies, including large-scale generic pretraining on an open-source facial dataset, to identify the most effective training regimes to enhance TED detection.

## II. RELATED WORK

### A. Deep Learning for Ophthalmic Disease and TED Detection

Deep learning has shown success across different medical disciplines. In ophthalmology, eye diseases are often diagnosed from imaging modalities such as fundus photographs or optical coherence tomography (OCT) scans. Babenko and colleagues [6] built an Inception-v3 based model to predict diabetic retinopathy (DR), diabetic macular edema, and poor blood glucose control using a multitask training objective. There have been an array of recent works using either orbital CTs or external facial images to classify TED.

Lin and colleagues [7] used orbital CTs to predict no TED, mild TED and severe TED with 1187 scans from 141 patients over a 10 year period and achieved high test accuracies. Wu and colleagues [8] also applied the Inception-v3 architecture on 55 normal and 3 abnormal cases on CT images. Ha and colleagues [9] applied VGG-16 to detect TED or orbital myositis on orbital CT images, with 1628 single coronal slices from 31 controls, 83 mild TED, 40 severe TED, and 51 orbital myositis patients. Karlin and colleagues [10] used an ensemble of models to achieve high TED prediction performance with a dataset of 2248 facial images in total. Successful TED detection in clinical settings requires data from multiple sites to ensure generalizability, motivating our use of FL to help integrate deep learning into the clinical workflow.

### B. Federated Learning in Medicine

Federated learning (FL) is a decentralized approach that enables training across multiple institutions while preserving privacy. This is particularly valuable in medicine where data privacy, security, and availability all pose significant challenges to centralized training and thus generalizability of model performance.

Dayan and colleagues [11] used data from 20 institutions to train a FL model that predicted future oxygen requirements of patients with COVID-19 using chest X-ray and electronic health record (EHR) features. The global FL model was more robust and achieved performance improvement at each site compared to local training. Pan and colleagues [12] applied personalized FL to EHR data to predict Alzheimer's disease progression, achieving an improvement in AUC compared to local models.

| Category | Attribute | TED | | CONTROL | |
|---|---|---|---|---|---|
| | | Columbia (N=135) | Stanford (N=100) | Columbia (N=156) | Stanford (N=100) |
| Sex | Female | 111 (82%) | 86 (86%) | 123 (79%) | 86 (86%) |
| | Male | 24 (18%) | 14 (14%) | 33 (21%) | 14 (14%) |
| Race | American Indian or Alaska Native | 0 (0%) | 0 (0%) | 0 (0%) | 0 (0%) |
| | Asian | 8 (6%) | 40 (40%) | 12 (8%) | 32 (32%) |
| | Black or African American | 16 (12%) | 2 (2%) | 10 (6%) | 0 (0%) |
| | Native Hawaiian or Other Pacific Islander | 0 (0%) | 4 (4%) | 0 (0%) | 0 (0%) |
| | White | 59 (44%) | 25 (25%) | 88 (56%) | 42 (42%) |
| | Other | 17 (13%) | 24 (24%) | 21 (13%) | 17 (17%) |
| | Declined | 35 (26%) | 5 (5%) | 25 (16%) | 9 (9%) |
| Ethnicity | Hispanic or latino/a | 15 (11%) | 16 (16%) | 20 (13%) | 12 (12%) |
| | Not hispanic or latino | 85 (63%) | 78 (78%) | 107 (69%) | 79 (79%) |
| | Declined | 35 (26%) | 6 (6%) | 29 (19%) | 9 (9%) |
| Conditions | Thyroid eye disease | 135 (100%) | 100 (100%) | – | – |
| | Eyelid Lesion | – | – | 56 (36%) | 0 (0%) |
| | Eyelid aging/malposition | – | – | 70 (45%) | 95 (95%) |
| | Epiphora | – | – | 40 (26%) | 0 (0%) |
| | Non-structural cosmetic issues | – | – | 5 (3%) | 5 (5%) |
| | Other | – | – | 11 (7%) | 0 (0%) |

**TABLE I:** Columbia/Stanford Data Demographics and Summary Statistics

In ophthalmology, Gholami and colleagues [13] [14] demonstrated the effectiveness of integrating self-supervised pretraining with domain-adaptation for federated learning of ophthalmic diseases using OCT images. They showed that applying this framework led to a significant increase in model performance compared to local models.

Given the sensitivity of facial images and the limited data availability, FL offers a compelling solution for collaboratively developing robust models while preserving patient privacy.

## III. METHODOLOGY

Our framework consists of a server orchestrating federated training with 2 clients for 10 rounds. Each round includes local training for 10 epochs for each client, respectively. Personalized FL occurs after global training, where the global model is finetuned on local data for 10 epochs. Local training involves either self-supervised pretraining or finetuning, and image preprocessing is applied before all model training. Figure 1 illustrates our pipeline.

### A. Data

**Data:** Table I presents summary statistics of datasets across different sites. Notably, the demographic distributions and conditions for controls vary between the two institutions in this federated collaboration: Columbia and Stanford. The Stanford dataset contains a higher proportion of eyelid aging/malposition, while the Columbia dataset includes more eyelid lesions and epiphora. Stanford also contains a much higher percentage of Asians compared to Columbia.

**Preprocesing:** To standardize inputs to our model, we crop images around the periocular region by first detecting corners of the eye and cropping the surrounding area with MediaPipe [15]. This preprocessing is applied to all data from all sites, including the external FFHQ dataset used for pretraining.

**Pretraining Data:** In addition to pretraining on patient data, we also consider pretraining on a large dataset of facial images to further improve representations. Flickr-Faces-HQ (FFHQ) [5] is a high quality dataset of human faces, originally curated for training generative adversarial networks. It contains 70,000 facial images of people across different age and ethnicity groups as well as from different backgrounds. The aligned and cropped images were used to pretrain our model, using the same preprocessing pipeline described in Figure 1. For the minority of images for which MediaPipe failed to detect key points of the eye, we manually cropped the images.

### B. Models

*1) Image Encoders:* We focus on ResNet-18 [16] and ViT-Base (ViT-B) [17] model backbones, and for all pretained ViTs, we utilize Masked Autoencoder (MAE)-pretrained ViTs. Empirically, we find that ResNet-50 can degrade performance due to overfitting on our relatively small local training datasets; therefore, ResNet-50 is excluded from our experiments. Ensemble of ResNet-18 [10] is also evaluated to compare with non-ensemble architecture for examining if variance reduction techniques could yield consistent performance gain.

*2) Self-Supervised Pretraining:* To improve learned representations and robustness of our model, we first pretrain our image encoders with self-supervision. We evaluate two different self-supervision techniques, MAE and SimCLR, to understand how different pretraining methods affect detecting TED. These methods represent different paradigms: MAE focuses on reconstruction which allows the model to capture global context from incomplete data, while contrastive objectives force the model to learn invariant representations.

**Masked Autoencoder:** Masked autoencoders (MAE) use vision transformers (ViT) as their backbone, and their pretext task is to predict masked-out patches' pixel values. A typical MAE consists of an asymmetrical transformer encoder, a ViT, and a transformer decoder. During pretraining, 75% of the patches are masked out, and only the unmasked patches are passed through the encoder. The encoded patches are then combined with masked tokens and decoded through the decoder. MAEs are trained with mean square error objective.

**SimCLR:** SimCLR is a contrastive self-supervision framework, where each image is passed through a set of augmen-

| Method | FFHQ | Columbia | Acc | Precision | Recall/Sensitivity | Specificity | F1-Score |
|---|---|---|---|---|---|---|---|
| Direct Finetuning* | - | - | 75.86% ± 12.90 | 71.92% ± 12.74 | 78.41% ± 17.42 | 73.71% ± 12.51 | 74.66% ± 14.44 |
| Direct Finetuning† | - | - | 78.23% ± 10.16 | 76.01% ± 11.44 | 78.96% ± 11.46 | 77.46% ± 12.75 | 77.14% ± 12.32 |
| Ensemble [10] | - | - | 76.55% ± 11.81 | 76.10% ± 14.61 | 74.56% ± 13.25 | 78.13% ± 18.26 | 74.76% ± 12.36 |
| MAE | - | ✓ | 77.19% ± 9.590 | 76.08% ± 12.89 | 72.49% ± 18.44 | 82.97% ± 8.720 | 76.08% ± 12.89 |
| MAE | ✓ | ✓ | 78.16% ± 8.260 | 79.48% ± 7.210 | 83.74% ± 7.65 | 72.58% ± 18.52 | 77.22% ± 9.710 |
| SimCLR | - | ✓ | 76.55% ± 10.23 | 75.04% ± 12.60 | 79.12% ± 12.32 | 74.21% ± 20.16 | 76.05% ± 9.24 |
| SimCLR | ✓ | ✓ | 77.24% ± 9.37 | 74.51% ± 11.29 | 81.92% ± 11.75 | 72.96% ± 18.56 | 77.09% ± 8.15 |

**(a) Columbia:** Local Performance on Columbia Dataset

| Method | FFHQ | Stanford | Acc | Precision | Recall/Sensitivity | Specificity | F1-Score |
|---|---|---|---|---|---|---|---|
| Direct Finetuning* | - | - | 89.50% ± 13.48 | 90.01% ± 12.95 | 88.00% ± 16.61 | 91.00% ± 10.44 | 93.71% ± 13.50 |
| Direct Finetuning† | - | - | 90.00% ± 6.82 | 90.33% ± 7.83 | 90.00% ± 7.75 | 90.00% ± 8.94 | 90.06% ± 6.89 |
| Ensemble [10] | - | - | 95.91% ± 5.14 | 93.45% ± 8.61 | 99.00% ± 3.0 | 92.00% ± 10.77 | 96.25% ± 4.58 |
| MAE | - | ✓ | 93.00% ± 8.12 | 94.14% ± 8.13 | 92.00% ± 9.80 | 94.00% ± 9.17 | 92.85% ± 8.31 |
| MAE | ✓ | ✓ | 93.33% ± 9.65 | 93.48% ± 10.09 | 93.33% ± 9.43 | 93.33% ± 10.27 | 93.37% ± 9.59 |
| SimCLR | - | ✓ | 88.00% ± 8.12 | 89.69% ± 7.90 | 86.00 ± 11.14 | 90.00% ± 7.75 | 87.55% ± 8.58 |
| SimCLR | ✓ | ✓ | 91.00% ± 8.89 | 93.14% ± 9.75 | 89.00% ± 10.44 | 93.00% ± 10.05 | 90.76% ± 8.96 |

**(b) Stanford**: Local Performance on Stanford Dataset

**TABLE II:** Performance comparison of training methods on Columbia and Stanford datasets. ∗ denotes ResNet-18 as backbone, and † denotes MAE pretrained ViT-B as backbone.

tations, creating two views of the same image, $x_i$ and $x_j$. The objective is to maximize the agreement of the latent representation, $z_i = f(x_i)$ and $z_j = f(x_j)$, since both $z_i$ and $z_j$ stem from the same image, while maximizing disagreement with latent representation of other samples in the batch. SimCLR relies heavily on augmentation, since it increases the diversity of different possible views to allow the encoders to learn similarities between two images. SimCLR minimizes the objective $\mathcal{L} = \mathbb{E}[\mathbf{l}(i,j) + \mathbf{l}(j,i)]$, where:

$$\mathbf{l}(i,j) = -\log \frac{\exp(z_i \cdot z_j / \tau)}{\sum_{a \in A(i)} \exp(z_i \cdot z_a / \tau)} \qquad (1)$$

$A(i)$ is the set of all latent embeddings except $x_i$, and $\tau$ is the temperature.

*3) Federated Learning:* Federated learning enables distributed privacy-preserving training; both self-supervised pretraining and supervised finetuning are trained in a federated manner. We adopt Federated Averaging [18] as our main federated algorithm. During pretraining, the server shares all meta-data including learning rate, epoch, model architecture, and batch size. Each client then trains for a set number of epochs and shares its weights with the server. The server then averages the weights and sends back the updated global model weights to the local clients.

We also consider the personalized FL approach, similar to that used by Pan and colleagues [12], to address the heterogeneity of the data distribution across the two different sites, whereby we finetune the global model to enhance local adaptation.

*4) Model Training:* We applied different data augmentations and regularization techniques to avoid overfitting.

**Data:** Images are resized to $512 \times 512$ for ResNet-18, and $224 \times 224$ for ViT. We found that ResNet does not converge with smaller input-image size; hence, a higher resolution is adapted to ensure stable training and convergence. Augmentations include random resize crop, rotation, flip and RandAugment [19] applied randomly.

| Method | FFHQ | Columbia | AUC |
|---|---|---|---|
| Direct Finetuning* | - | - | 83.38% ± 11.87 |
| Direct Finetuning† | - | - | 86.54% ± 8.60 |
| Ensemble [10] | - | - | 83.70% ± 9.68# |
| MAE | - | ✓ | **88.35% ± 5.37** |
| MAE | ✓ | ✓ | 82.80% ± 8.16 |
| SimCLR | - | ✓ | 82.33% ± 6.78# |
| SimCLR | ✓ | ✓ | 86.30% ± 6.78 |

| Method | FFHQ | Stanford | AUC |
|---|---|---|---|
| Direct Finetuning* | - | - | 93.30% ± 11.83# |
| Direct Finetuning† | - | - | 96.70% ± 3.49# |
| Ensemble [10] | - | - | **99.70% ± 0.46** |
| MAE | - | ✓ | 97.10% ± 3.88 |
| MAE | ✓ | ✓ | 97.17% ± 4.63 |
| SimCLR | - | ✓ | 94.80% ± 5.06# |
| SimCLR | ✓ | ✓ | 95.80% ± 4.53# |

**TABLE III:** Area-under-the-curve (AUC) results on Columbia (top) and Stanford (bottom) datasets. *ResNet-18 backbone, †ViT-B backbone. #Statistically significant increase for Personalized FL MAE. See Table IV.

**Optimization:** We use AdamW [20] optimizer with weight decay of $1e-5$ for both pretrainined and supervised finetuning. For self-supervised pretraining, we use learning rate of $2e-5$ with 5 warm-up epochs with $1e-2$ rate. We pretrain for 30 epochs on FFHQ, 50 epochs on local datasets, and 50 epochs for supervised finetuning. We employ a cosine annealing scheduler with $1e-5$ minimum rate. We use early stopping with validation loss to prevent overfitting.

To avoid client drift during FL training, each round would only consist of 10 epochs, and 10 FL rounds are performed.

## IV. RESULTS

In this section, we show that our framework enables effective detection of TED by leveraging FL. We divide our experiments into two stages: local and FL training.

Local non-FL experiments confirm MAE consistently outperforms direct finetuning with highest AUC of 88.35% on Columbia data and AUC of 97.10% on Stanford data, as well as lowest variance on both datasets (excluding the ResNet

| Method / *Test Set* | Acc | Precision | Recall | Specificity | F1-Score | AUC |
|---|---|---|---|---|---|---|
| Direct Finetuning* | | | | | | |
| Global | | | | | | |
| *Columbia* | 74.83% ± 8.31 | 73.17% ± 7.80 | 71.59% ± 17.38 | 77.46% ± 7.63 | 71.57% ± 11.55 | 82.76% ± 9.82♯ |
| *Stanford* | 74.50% ± 11.50 | 69.33% ± 13.88 | 97.00% ± 4.58 | 52.00% ± 24.41 | 79.98% ± 8.15 | 95.30% ± 5.50♯ |
| Personalized | | | | | | |
| *Columbia* | 76.55% ± 10.99 | 74.55% ± 11.35 | 73.68% ± 21.18 | 78.75% ± 9.64 | 73.13% ± 14.94 | 85.05% ± 10.18♯ |
| *Stanford* | 92.50% ± 7.50 | 92.64% ± 10.49 | 93.00% ± 7.81 | 92.00% ± 14.00 | 92.88% ± 6.72 | 96.80% ± 3.82 ‡♯ |
| Direct Finetuning† | | | | | | |
| Global | | | | | | |
| *Columbia* | 76.55% ± 4.57 | 72.44% ± 5.12 | 80.50% ± 14.54 | 72.83% ± 8.89 | 75.50% ± 7.73 | 84.62% ± 6.77♯ |
| *Stanford* | 79.50% ± 11.06 | 74.57% ± 11.92 | 95.00% ± 8.06 | 65.00% ± 20.13 | 82.59% ± 8.53 | 95.50% ± 5.02♯ |
| Personalized | | | | | | |
| *Columbia* | 82.41% ± 4.57 | 80.80% ± 6.78 | 82.53% ± 6.62 | 82.62% ± 7.11 | 81.30% ± 5.21 | 88.56% ± 5.77 ‡ |
| *Stanford* | 93.00% ± 6.78 | 93.65% ± 8.40 | 93.00% ± 7.81 | 93.00% ± 10.05 | 93.06% ± 6.68 | 97.50% ± 3.00 ‡ |
| MAE | | | | | | |
| Global | | | | | | |
| *Columbia* | 78.62% ± 7.20 | 81.81% ± 11.55 | 71.65% ± 14.06 | 75.15% ± 9.08 | 84.67% ± 10.98 | 85.47% ± 9.06 |
| *Stanford* | 85.50% ± 7.57 | 82.13% ± 11.94 | 95.00% ± 9.22 | 76.00% ± 18.00 | 87.06% ± 6.17 | 97.30% ± 3.20 |
| Personalized | | | | | | |
| *Columbia* | 81.72% ± 9.39 | 82.51% ± 13.82 | 79.84% ± 13.33 | 83.50% ± 14.67 | 79.99% ± 10.92 | 89.26% ± 6.99 |
| *Stanford* | 94.50% ± 5.22 | 95.42% ± 5.92 | 94.00% ± 9.17 | 95.00% ± 6.71 | 94.32% ± 5.68 | 98.70% ± 3.26 ‡ |

**TABLE IV:** Federated training performance across different methods and backbones. We present global federated model as well as personalized model. Personalized models are finetuned on local data and then tested on the datasets shown. ∗ denotes ResNet-18, † denotes ViT, and ‡ denotes statistically significant increase in AUC compared to the global counterpart using DeLong's test, where $p < 0.05$. ♯ denotes statistically significant increase for Personalized FL MAE at corresponding site.

ensemble). Although the ensemble model offers improvement for Stanford data, we focus on single-model improvements to isolate the benefit of FL and pretraining.

We show that personalized FL MAE achieves AUC of 89.26% vs. 88.35% via local MAE for Columbia and 98.70% vs. 97.10% for Stanford, as shown in Table IV and Figure 2.

We test our model with collaborative cross-validation, inspired by [21], through which each institution's data is split into $k$-folds, and fold $i$ from both sites are paired for validation. All results we present are 10-fold cross-validated.

## A. Local Site Training Performance

Table III reports area under the curve (AUC), and table II presents accuracy, precision, recall/sensitivity, specificity, and F1 score for local site performance.

*1) Direct Finetuning:* Using ImageNet-pretrained ResNet-18 and ViT backbones, ViT consistently outperforms ResNet-18 across both sites. On Columbia data, ViT achieves 78.23% accuracy, 86.54% AUC (3.16% improvement over ResNet-18 AUC), improved precision, recall, specificity, and F-1 score. Local performance on Stanford's dataset also shows similar trends, with higher AUC and reduced variance, 96.70% ($\sigma = 3.49$), for ViT compared to ResNet-18's AUC of 93.30% ($\sigma = 11.83$).

These results suggest that ViT, benefiting from its transformer-based architecture, model size, and strength of MAE pretext tasks, may be better suited at capturing subtle visual features relevant to TED detection from external facial images even with smaller input images. Furthermore, although an ensemble of 5 ResNet-18 models attains the highest Stanford AUC, its Columbia performance matches that of a single ResNet-18; we therefore don't consider ensemble models for our FL runs.

*2) Self-Supervised Pretraining Improves Performance:* We compare different pretraining methods and varying amounts of pretraining data. For both MAE and SimCLR, we evaluate two strategies: (1) pretraining first with FFHQ followed by local site dataset, and (2) pretraining on local site dataset alone. The goal is to assess whether pretraining on a large, diverse facial-images dataset improves our model's ability to learn subtle differences for the downstream TED detection task. SimCLR uses ResNet-18 as backbone, while MAE uses ViT-B as backbone.

MAE-pretrained ViT offers slightly better results, with higher precision achieved by both MAE-pretrained models compared to SimCLR-pretrained models at both sites. Notably, MAE pretraining on both Columbia's and Stanford's dataset exhibits the highest AUC and lowest standard deviation (after the ResNet18 ensemble for Stanford). MAE pretraining on FFHQ followed by pretraining (for Columbia only), on the other hand, does not yield meaningful improvement, and this is most likely due to our use of an ImageNet MAE-pretrained ViT. This suggests ImageNet potentially provides sufficient pre-training in spite of not being a tailored facial-image dataset. Interestingly, recall and specificity are flipped between MAE pretraining with Columbia data only vs. MAE pretraining with FFHQ pretraining followed by Columbia pre-training. This is potentially due to non-converging validation folds and the inherent randomness in each validation run. In these cases, models may predict near-random probabilities with a slight class bias. Therefore, recall and specificity across all methods tend to show similar mean and standard deviation or skew towards one class. This behavior is not seen on the Stanford dataset, as all Stanford methods achieve very high performance compared to their Columbia counterparts, suggesting Stanford's dataset has patterns that our models are

able to distinguish more easily overall.

## B. Federated Training Performance

We compare federated training with and without self-supervised pretraining. We found that personalized FL is paramount to achieving good performance due to data availability and distribution shift. Our main comparison for FL is between models with FL, shown in Table IV, and without FL, shown in Tables IIa and IIb along with AUCs in Tables III. We show that all FL models achieve comparable or better performance than local models.

*1) Direct Finetuning:* Personalized FL ResNet-18 achieves higher AUC of $85.05\%$ on Columbia's dataset (compared to local AUC shown in Table III); it also achieves higher AUC of $96.80\%$ compared to the local AUC on Stanford's dataset (shown in Table III). Direct finetuning alone shows promising results, with personalized FL ViT showing improvement compared to local models on both Columbia and Stanford datasets. For Columbia, personalized FL ViT shows an improvement across all metrics with accuracy of $82.41\%$ compared to $78.23\%$ without FL training (Table IIa), and it does not suffer from validation instability, as seen by reduced variance in recall and specificity. On Stanford's dataset, we also see a small improvement for the personalized FL ViT over its non-FL model (Table IV).

*2) MAE:* MAE pretraining helps improve personalized model performance on both client sites. On Columbia data, there is a slight improvement in accuracy and AUC of $89.26\%$ compared to $88.35\%$ on local data only. On Stanford's data, FL MAE-pretraining exhibits an improvement across all metrics compared to other personalized models and all local models, excluding the ensemble model, achieving single network AUC of $98.70\%$. Pairwise DeLong's tests show that Personalized FL MAE models have higher AUC ($p < 0.05$) than 5/12 and 8/12 models for Columbia and Stanford, respectively, including local and FL models. This indicates that FL and specifically MAE-pretraining do improve overall model performance.

## C. Limitations

*1) Limited Institutional Diversity:* Our federated framework is evaluated on two institutions, which might not capture all exogenous TED variations, including age and ethnic distribution shifts. This also limits the amount of training data available, therefore hindering model performance. However, we expect model performance to improve using our proposed framework with the addition of more sites, and future work will scale up the number of participating institutions to capture the spectrum of TED across broader populations.

*2) Validation Instability in Limited-Data Regimes for Local Sites:* Despite improved robustness through self-supervised training, we observed high variance in model performance across validation folds in some settings, particularly in folds where training data did not align well with the validation set distribution. We qualitatively examined these folds containing a disproportionately higher number of mild TED cases. Such cases exhibit subtle features, making them challenging to detect. This challenge can be mitigated by expanding the local datasets at each site and increasing the number of sites to improve representation of mild cases. Future work toward domain adaptation between local sites could address this by applying domain adaptation techniques, as shown in [13].

## V. Discussion

### A. Role of Pretraining, Importance of FL, Feature Analysis

In Section IV-A, we presented that MAE-pretrained ViT on both datasets has the highest cross-validated AUC among the different models for local training. We also found that additional facial-image pretraining exhibited comparable performance to ImageNet pretraining alone. At the same time, the performance gain from SSL-pretraining was marginal; performance across methods and backbones saturates for local models, regardless of SSL-pretraining. We hypothesize that this is due to the heterogeneity of TED manifestations and the lack of representation of these variations at a given local site. Across different cross-validation folds and across methods, there are folds where test loss does not converge due to differences between training and testing sets, further emphasizing the need for federated training. We conducted analysis of facial features used by our models by examining attention maps of MAE-pretrained ViT and found that the model frequently focused on the palpebral fissure height (the vertical distance between the upper and lower eyelid margins when the eyes are open) as well as the eyebrow region. These areas are clinically meaningful and consistent with existing literature.

### B. Importance of Personalization in FL for TED

Our experiments showed that personalization in FL is important, especially when the data distribution between two sites is drastically different. This approach combined the benefits from both collaborative training and specializing to local data distributions. We performed Delong's test to compare AUCs and found that personalized FL ViT on both sites, FL MAE on Stanford data, and FL ResNet-18 on Stanford data, all exhibited statistically significant performance improvement ($p < 0.05$) compared to global FL models.

Collaborative training for TED, especially with heterogeneous data distributions, requires personalization to achieve the best results.

## VI. Conclusions and Future Directions

In this study, we developed and evaluated a framework for federated learning of deep learning based detection of thyroid eye disease. We integrated self-supervised learning into model training to improve model representations. We found that MAE pretraining outperforms SimCLR pretraining and that additional self-supervised pretraining on a large facial-image dataset was not needed to achieve maximal performance on datasets at local sites. While these findings streamline local training pipelines, they also emphasize the need for cross-institution collaboration to develop high-performing deep learning models for TED detection. The future vision

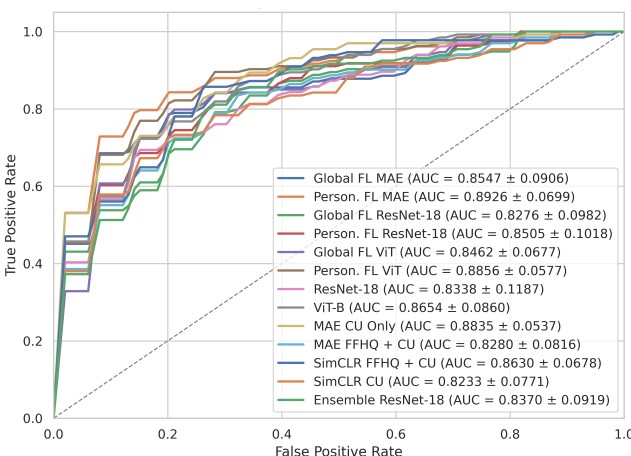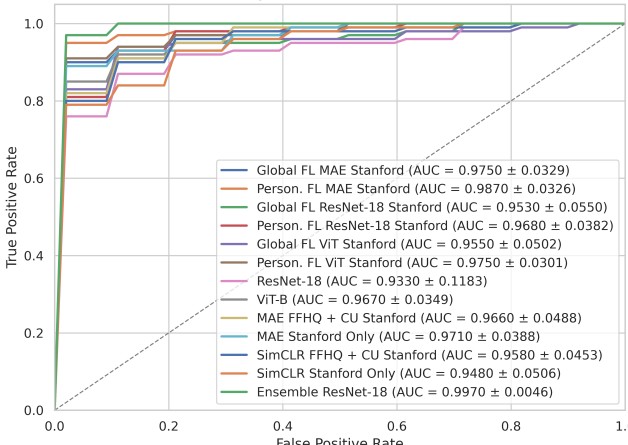

**Fig. 2:** Receiver operating characteristic (ROC), on Columbia (left) and Stanford (right), trained locally and via FL.

for our study is to ultimately deploy our TED models on smartphones, enabling federated training using images captured from distributed smartphone cameras to expedite early TED detection. To further validate the clinical impact of our workflow, a third institution will be added as a held-out validation set; we are also expanding the number of data samples for training at each site for future iterations of our algorithm. We anticipate our MAE approach will exhibit the best generalizability due to MAE pretraining. Different FL algorithms such as SCAFFOLD [22] can also be implemented in future work to address differences in data distributions across sites, potentially adding communication overhead to the FL process while improving performance. Lastly, our study lays the groundwork for future cross-institutional collaboration through FL using sensitive medical data, incorporating multimodal information such as patient history, genetic data, and other imaging modalities, moving towards personalized medicine.

## Acknowledgment

The authors thank Xue (Mia) Dong (Stanford) for her instrumental help in data collection and Theodore Leng (Stanford) and Eric N. Brown (Vanderbilt) for their valuable insights.

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
