# OpenReview forum: "FedTED: Federated Learning for Robust Thyroid Eye Disease Detection with Masked Autoencoders"
_IEEE.org/EMBS/BHI/2025/Conference — BHI 2025_

### Official Review · Reviewer_DN9Q · 2025-06-27
**FedTED: Federated Learning for Robust Thyroid Eye Disease Detection with Masked Autoencoders**

**Confidence:** 3
**Clarity Of Writing:** good
**Clinical Significance:** good
**Methodological Novelty:** good
**Overall Rating:** 4
**Final Rating:** 7

**Experiments And Results:**

good

**Questions For The Authors:**

Questions For The Authors

Can you provide more details on the dataset split and diversity across sites? How were differences in demographics, lighting, or imaging conditions handled?

Have you compared the performance of FedTED to clinical experts or traditional clinical workflows for TED diagnosis?

Was communication cost or training efficiency considered in the federated setting? If not, what would be the expected impact on scalability?

Could you provide an ablation study breaking down the impact of MAE, contrastive pretraining, and federated fine-tuning separately?

Are there plans for external validation using datasets not included in the federated training setup?

**Strengths:**

Strengths

Privacy-preserving approach: Use of federated learning ensures sensitive patient data remain local, which is critical in medical applications.

Strong empirical results: Achieves high AUC (98.70%) across validation folds, indicating robustness and effectiveness.

Timely problem: Addresses data sharing challenges in healthcare, especially relevant to rare or less well-represented diseases like TED.

Good integration of techniques: Combines MAEs and contrastive pretraining in a federated learning setting, which is innovative and shows thoughtful methodology.

Clear clinical application: Directly applicable to non-invasive facial image analysis, which is promising for early screening of TED.

**Summary Of The Paper:**

This paper presents FedTED, a novel framework that combines federated learning, self-supervised pretraining with masked autoencoders (MAE), and contrastive learning for detecting Thyroid Eye Disease (TED) from external facial images. The approach is privacy-preserving and aims to address the challenges of data scarcity and institutional data silos by avoiding raw data sharing. The authors evaluate FedTED across multiple training regimes and report superior AUC performance (up to 98.70%) compared to supervised baselines. The application is clinically relevant and the framework is positioned for deployment in real-world diagnostic settings.

**Weaknesses:**

Weaknesses

Limited information on dataset characteristics: It is unclear how diverse or representative the training data are across institutions, which affects generalizability.

Lack of ablation studies: The individual contributions of federated learning, MAE, and contrastive learning are not dissected. A detailed ablation would clarify what contributes most to performance gains.

Unclear clinical validation: While high AUC is promising, there is little discussion on real-world deployment or comparison to clinical expert performance.

Scalability and communication overhead in federated learning are not addressed, which can be important for deployment across multiple clinical sites.

---

### Official Review · Reviewer_Jr3K · 2025-07-04
**Review on "FedTED: Federated Learning for Robust Thyroid Eye Disease Detection with Masked Autoencoders"**

**Confidence:** 5
**Clarity Of Writing:** good
**Clinical Significance:** great
**Methodological Novelty:** good
**Overall Rating:** 5

**Experiments And Results:**

good

**Questions For The Authors:**

### **1. Absence of Centralized Training Results**

**Potential misunderstanding**: I assume centralized (pooled) training was not performed or not reported. However, it's possible that the authors ran such experiments but excluded them for space or clarity reasons.

**Questions**:

* Did you train any models using pooled data from site-1 and site-2 (i.e., a centralized setting)?
* If so, how does the performance compare to federated or local models, particularly the personalized FL models?

---

### **2. Backbone Confounding in MAE vs. SimCLR Comparisons**

**Potential misunderstanding**: The paper compares MAE (ViT) and SimCLR (ResNet-18) without matching backbones. I interpret the observed performance differences as being partly due to architectural differences, not just self-supervised method.

**Questions**:

* Did you consider controlling for architecture by running both MAE and SimCLR with the same backbone (e.g., ViT)?
* If not, can you explain why this comparison is valid or necessary as is?

---

### **3. Details of Federated Training Protocol**

**Potential misunderstanding**: It is not fully clear how federated rounds were performed. I understand that each client performs 10 epochs per round for 10 rounds, but it is not stated:

* Whether model weights were reinitialized per round?
* Whether personalization was performed on frozen or fine-tuned backbones?
* How validation was done during FL—centrally or locally?

**Questions**:

* Can you confirm whether all models started from the same initialization (e.g., ImageNet or MAE weights) and were updated via FedAvg?
* How was personalization implemented? Did you fine-tune the global model further on each site's local data?
* Were validation metrics calculated locally on each site’s validation set or globally?


---

### **4. Use of FFHQ vs. ImageNet Pretraining**

**Potential misunderstanding**: The manuscript states that FFHQ pretraining did not improve performance much, but also notes that ImageNet MAE weights were used. It is unclear how the benefits of FFHQ pretraining were isolated.

**Questions**:

* Did you start MAE pretraining on FFHQ from scratch or from ImageNet-pretrained MAE weights?
* Could FFHQ's limited benefit be due to being a second-stage pretraining after ImageNet?
* Would FFHQ have more impact if used for initial MAE training?

---

### **5. Statistical Testing and Result Significance**

**Potential misunderstanding**: The paper claims statistical significance using Delong’s test but only annotates significant results with symbols (‡) without showing full pairwise comparisons.

**Questions**:

* Can you provide a full table of p-values for AUC comparisons (e.g., local vs. FL, SimCLR vs. MAE, personalized vs. global)?
* How many of the observed improvements are statistically significant across folds?

---

### **6. Cross-Site Generalization Evaluation**

**Potential misunderstanding**: I assume no experiment was done where a model trained on one site is tested on the other without fine-tuning.

**Questions**:

* Did you perform any cross-site generalization experiments (e.g., training on site-1, testing on site-2)?
* If not, do you believe such evaluation would yield lower performance than FL?

**Strengths:**

The manuscript demonstrates several **promising aspects** that make it a meaningful contribution to the fields of federated learning, medical AI, and ophthalmic disease detection:

---

### **1. Integration of Federated Learning with Self-Supervised Learning (MAE)**

* **Novelty**: The work presents *FedTED*, a framework that uniquely combines **federated learning (FL)** with **masked autoencoder (MAE) pretraining** for a medical application. This is, to the best of current knowledge, the **first application of FL + MAE for TED detection**, which addresses both privacy and data scarcity issues in healthcare.
* **Impact**: This combination allows institutions to collaboratively train robust models **without sharing sensitive patient data**, addressing critical ethical and logistical barriers in cross-institutional AI development.

---

### **2. Application to a Clinically Significant and Understudied Problem**

* **Relevance**: Thyroid Eye Disease (TED) is a visually manifesting autoimmune condition where early detection is crucial, yet challenging due to its heterogeneous presentation.
* **Innovation**: The focus on **external facial images** (as opposed to CT or MRI) enhances the **accessibility and practicality** of the solution—enabling potential deployment in telemedicine and mobile diagnostics.

---

### **3. Demonstrated Effectiveness of Personalized FL**

* The study shows that **personalized federated learning** significantly improves performance compared to global FL models, especially in heterogeneous data settings.
* **Quantitative results** show statistically significant improvements (up to **98.70% AUC**) using personalized MAE-based models, which are competitive with or better than strong baselines including ensembles and SimCLR.

---

### **4. Robust Experimental Evaluation**

* The authors provide extensive **10-fold cross-validation**, multiple model architectures (ResNet-18, ViT-B), and **site-wise breakdown** of performance metrics (AUC, accuracy, precision, recall, specificity, F1-score).
* Comparative analysis across **MAE vs. SimCLR**, **local vs. federated**, and **personalized vs. global** FL models shows thoughtful investigation into what contributes to performance gains.

---

### **5. Practical Design Choices for Medical AI**

* **Data preprocessing** using the periocular region helps standardize input features, which improves model robustness across clinical sites.
* The use of **FFHQ pretraining** and **publicly available ViT models** helps reduce reliance on large, labeled medical datasets, addressing data scarcity common in healthcare ML.

---

### **6. Scalability and Deployment Potential**

* The authors explicitly discuss the possibility of **deploying their models on smartphones**, which aligns with current trends toward **edge-based federated inference** and **global-scale early disease detection**.
* The lightweight use of ViT-B and ResNet-18 suggests **computational feasibility** for real-world applications.

**Summary Of The Paper:**

The paper introduces **FedTED**, a framework that combines **federated learning (FL)** and **self-supervised learning**—specifically, **masked autoencoders (MAE)**—to detect **Thyroid Eye Disease (TED)** using external facial photographs. The primary goal is to enable **privacy-preserving**, **multi-institutional training** of machine learning models for TED detection without sharing sensitive patient data.

### Key Components:

1. **Federated Learning**:

   * Used to train models across two medical institutions without transferring raw image data.
   * Includes both **global** and **personalized** federated training approaches.

2. **Self-Supervised Pretraining**:

   * Models are pretrained using MAE and SimCLR on either:

     * Flickr-Faces-HQ (FFHQ) facial dataset
     * Local site-specific clinical data
   * MAE uses Vision Transformer (ViT-B) backbones; SimCLR uses ResNet-18.

3. **Model Training and Architecture**:

   * Two primary backbones are used: ResNet-18 and ViT-B.
   * Fine-tuning is conducted after self-supervised pretraining.
   * Performance is evaluated with and without federated training.

4. **Evaluation**:

   * 10-fold cross-validation on datasets from two institutions.
   * Comparison across direct finetuning, ensemble methods, SimCLR, and MAE.
   * Metrics: AUC, accuracy, precision, recall, specificity, and F1-score.
   * Best results: **AUC up to 98.70%** with personalized federated MAE on site-2 data.

5. **Findings**:

   * Self-supervised MAE pretraining improves performance over direct finetuning and SimCLR.
   * Personalized federated models outperform global FL models and local-only training.
   * FFHQ-based pretraining offers limited additional benefit beyond ImageNet MAE weights.

6. **Limitations Acknowledged**:

   * Only two sites are included, limiting generalizability.
   * High performance variance across folds in small data regimes.
   * Absence of centralized training for comparison.

### Conclusion:

FedTED effectively integrates FL and self-supervised learning for TED detection, achieving high accuracy while preserving data privacy. The study suggests the method is well-suited for broader deployment in real-world, privacy-sensitive medical contexts.

**Weaknesses:**

The current manuscript presents a strong foundation—combining federated learning (FL) with self-supervised masked autoencoders (MAE) for the clinically meaningful task of Thyroid Eye Disease (TED) detection. However, as submitted, the work **falls short of the publication bar primarily due to missing comparative baselines and insufficient analysis of key modeling assumptions**. Below, I outline specific directions and experimental additions that can substantially strengthen this work and potentially elevate it to a publishable standard:

---

### **1. Add Centralized Training Baselines**

**Why:** Centralized (pooled data) training represents the theoretical performance upper bound in a multi-institutional setting. Without it, we cannot assess the performance trade-off imposed by federated learning.

**What to do:**

* Train all model variants (ResNet, ViT, MAE-pretrained) using pooled data from site-1 and site-2.
* Compare AUC, accuracy, and variance to both personalized and global FL models.

**What it will show:**

* Whether FedTED approaches or exceeds centralized model performance.
* How much performance is being sacrificed (if at all) for privacy-preserving FL.

---

### **2. Include Cross-Site Generalization Experiments**

**Why:** A major claim of the paper is improved generalizability via FL. However, the current experiments do not test generalization across unseen domains (institutions).

**What to do:**

* Train a model on site-1, test on site-2 (and vice versa), with no fine-tuning.
* Optionally test global FL models on out-of-distribution site data.

**What it will show:**

* How well learned representations transfer across domains.
* Whether federated training improves cross-site robustness compared to local-only models.

---

### **3. Control for Backbone Differences in MAE vs. SimCLR**

**Why:** MAE uses ViT while SimCLR uses ResNet-18, making it unclear whether performance differences are due to the self-supervised method or the backbone architecture.

**What to do:**

* Run SimCLR with a ViT backbone or MAE with a lightweight ResNet-style transformer (e.g., DeiT or ViT-S).
* Keep training settings (augmentation, epochs, image resolution) constant.

**What it will show:**

* Whether MAE truly outperforms SimCLR due to its pretext task or if the ViT backbone alone drives the improvement.

---

### **4. Add Visualization of Learned Representations**

**Why:** The paper asserts that self-supervised learning improves feature quality, but provides no evidence to support this claim beyond performance metrics.

**What to do:**

* Use t-SNE or UMAP to project final-layer embeddings from local, global FL, and personalized FL models (across MAE and SimCLR).
* Color by class (TED vs. control) and by site (to reveal domain shifts).

**What it will show:**

* Whether MAE results in better class separation or site-invariant features.
* Qualitative evidence of representation robustness and domain adaptation.

---

### **5. Evaluate Clinical Utility Through Operating Threshold Analysis**

**Why:** Clinical decisions depend on sensitivity/specificity trade-offs, not just AUC. A high AUC model may still be unusable if false positives are too high.

**What to do:**

* Plot precision-recall curves and decision curves.
* Report optimal threshold selection strategies (e.g., Youden's J statistic) and associated metrics.

**What it will show:**

* How the model performs at clinically relevant thresholds.
* Whether personalization helps maintain specificity/sensitivity balance in practice.

---

### **6. Extend Dataset or Simulate Larger FL Setting**

**Why:** FL’s key strength is in scaling to many clients. Two institutions is a narrow setting.

**What to do:**

* Either add a third dataset (e.g., public dataset of facial conditions) as a simulated FL client, or simulate non-IID clients by splitting site-1 by demographic groups.

**What it will show:**

* The scalability of FedTED to more realistic multi-institutional or federated edge-device settings.
* Whether FedTED’s performance gains hold in higher heterogeneity scenarios.

---

### **7. Perform Error Analysis**

**Why:** The paper lacks any insight into where or why models fail. Understanding errors can guide improvements.

**What to do:**

* Sample and qualitatively analyze misclassified cases (especially false negatives).
* Stratify performance by subgroups (e.g., age, ethnicity, severity of TED).

**What it will show:**

* Whether the model has performance disparities.
* Which cases are most difficult and whether FL helps mitigate these challenges.

---

### Official Review · Reviewer_Buyv · 2025-07-06
**FedTED Some comments and suggestions**

**Confidence:** 5
**Clarity Of Writing:** great
**Clinical Significance:** great
**Methodological Novelty:** great
**Overall Rating:** 7
**Final Rating:** 7

**Experiments And Results:**

great

**Questions For The Authors:**

Q1: Privacy and Security: Did you empirically evaluate privacy risks (such as membership inference attacks) in the federated setting? Have you considered implementing more advanced privacy-enhancing mechanisms?

Q2: Communication Overhead: What are the actual communication and computational costs per round of federated learning? Is this method practical for resource-constrained environments?

Q3: Statistical Significance: Are the reported improvements in metrics such as AUC supported by p-values or confidence intervals?

Q4: Ablation on Pretraining Data: Since pretraining with FFHQ yielded limited improvement, is it necessary to invest in large-scale domain-specific pretraining?

Q5: Open-Source Plans: Will the code, models, or data preprocessing pipelines be made publicly available?

Q6: Clinical Impact: With improved sensitivity and specificity through federated learning, how would actual patient screening or referral workflows be affected in practice?

**Strengths:**

S1: This is the first work to apply federated masked autoencoder pretraining for TED detection from facial images, combining state-of-the-art federated learning and self-supervised vision transformer methods.

S2: The approach addresses a real-world bottleneck: how to build robust models for rare diseases from geographically and demographically diverse populations under strict privacy constraints.
S3: The experiments include comparisons with multiple pretraining strategies (MAE, SimCLR), different backbone architectures (ResNet-18, ViT), both local and federated training, and a range of metrics (accuracy, AUC, F1, etc.), across two clinical sites with distinct patient distributions.

S4:  Federated MAE-based models consistently outperform both local and non-pretrained baselines, with personalized federated models achieving up to 98.7% AUC.

**Summary Of The Paper:**

This manuscript proposes FedTED, a privacy-preserving federated learning framework that integrates self-supervised masked autoencoder (MAE) pretraining for robust Thyroid Eye Disease (TED) detection from external facial images. The method enables collaborative multi-site model training without sharing sensitive patient data. The authors evaluate the approach on multi-institution datasets and show that federated MAE-based models outperform supervised baselines and achieve high AUC, demonstrating the potential of combining federated learning with self-supervised training for privacy-sensitive clinical applications.

**Weaknesses:**

W1: There is considerable variation in model performance across validation folds, especially in folds where the distribution of training and validation data is not well matched. The authors should briefly analyze and clarify whether this is mainly due to inter-individual differences or device acquisition differences, and discuss potential approaches for improvement.

W2: Although the model achieves high accuracy, the impact on real clinical workflows and patient outcomes is not discussed in sufficient depth.

W3: The paper lacks detailed experiments or discussion regarding the core issues of communication overhead and privacy risks in federated learning.

---

### Official Review · Reviewer_Fw6j · 2025-07-15
**Review for FedTED: Federated Learning for Robust Thyroid Eye Disease Detection with Masked Autoencoders**

**Confidence:** 3
**Clarity Of Writing:** good
**Clinical Significance:** good
**Methodological Novelty:** great
**Overall Rating:** 7
**Final Rating:** 7

**Experiments And Results:**

great

**Questions For The Authors:**

* Given the validation instability you observed, how would you ensure reliable performance in clinical deployment?
* Can you provide analysis of what facial features the MAE-pretrained models focus on for TED detection? This would help validate clinical relevance and build trust with clinicians.
* How many federated learning rounds are actually needed for convergence? and why 10 rounds is selected for training FL model?
* Could you clarify exactly when and how personalization occurs? Are there additional epochs used?

Answering these questions may help address the concern in the weakness section.

**Strengths:**

The paper addresses a genuine medical need where early TED detection could significantly impact treatment outcomes. The integration of federated learning with self-supervised pretraining (MAE) for medical imaging represents a meaningful technical contribution for medical applications where data privacy is important and datasets are limited. Although the data set is small, it represented the diversity of the population. The paper includes systematic comparisons across different architectures (ResNet-18 vs ViT), pretraining strategies (MAE vs SimCLR, with and without FFHQ), and training paradigms (local vs federated vs personalized), reflecting comprehensive thoughts in the study design. Overall, this is a pretty strong paper, with some minor concerns that can make it more robust.

**Summary Of The Paper:**

The paper recognizes the need for early and quick detection of Thyroid Eye Disease (TED) and the challenges in developing robust systems from limited datasets with confidentiality concerns. Hence, it proposes FedTED, a privacy-preserving framework that combines federated learning with self-supervised pretraining (Masked Autoencoders) for TED detection from facial photographs. This approach enables collaboration between different medical institutions without sharing sensitive patient data. The authors evaluate their framework across two clinical sites with different patient demographics and demonstrate that federated MAE-based models outperform supervised baselines, achieving up to 98.70% AUC. The work addresses an important clinical need for early TED detection while respecting privacy constraints inherent in medical data sharing.

**Weaknesses:**

**Clarity concerns:**

* In the related work section, the paper introduced the previous studies using deep learning and federated learning, but it would be more clear to include a concluding sentence under each subsection on the research gaps that motivated the current study.

* The methods involve novel and complex combinations that require clearer illustration. Figure 1a should explicitly show the privacy-preserving aspect by illustrating that only model weights (not patient data) cross institutional boundaries, (e.g. Consider adding a "data stays local" annotation) Additionally, Figure 1 can benefit more by providing a more comprehensive schematic illustrating all encoder/pretraining combinations evaluated in the study to help readers understand the approach.

* The current FL description lacks detail about the differences between personalized FL versus global FL approach. The "when and how" personalization occurs in the training is needed.

**Methodology concerns:**
* Lack of justification on the FL training rounds. The 10 rounds sound arbitrary without convergence analysis. Please explain the reason for the selection of 10 epochs in the text.

**Clinical relevance discussion:**

* While performance metrics are strong, there is little discussion on what image features the model is learning (e.g., feature importance, attention map). The interpretability analysis would help improve the clinical application and usability of the model.

* The authors acknowledge high variance across validation folds and non-converging cases, suggesting potential overfitting issues. In the evaluation framework, external validation on an independent institution is lacking. It would be better if the authors can find data from a third institution to examine the generalizability. Given that data might be limited, the paper should at least comment more on the generalizability of their model in the discussion.